# A Comparative Study of Neurosymbolic AI Approaches to Interpretable Logical Reasoning

**Michael K. Chen**                                              MICHAELCHENKJ@GMAIL.COM

*Raffles Institution*

**Editors:** Leilani H. Gilpin, Eleonora Giunchiglia, Pascal Hitzler, and Emile van Krieken

## Abstract

General logical reasoning, defined as the ability to reason deductively on domain-agnostic tasks, continues to be a challenge for large language models (LLMs). Current LLMs fail to reason deterministically and are not interpretable. As such, there has been a recent surge in interest in neurosymbolic AI, which attempts to incorporate logic into neural networks. We first identify two main neurosymbolic approaches to improving logical reasoning: (i) the **integrative approach** comprising models where symbolic reasoning is contained within the neural network, and (ii) the **hybrid approach** comprising models where a symbolic solver, separate from the neural network, performs symbolic reasoning. Both contain AI systems with promising results on domain-specific logical reasoning benchmarks. However, their performance on domain-agnostic benchmarks is understudied. To the best of our knowledge, there has not been a comparison of the contrasting approaches that answers the following question: Which approach is more promising for developing general logical reasoning? To analyze their potential, the following best-in-class domain-agnostic models are introduced: Logic Neural Network (LNN), which uses the integrative approach, and LLM-Symbolic Solver (LLM-SS), which uses the hybrid approach. Using both models as case studies and representatives of each approach, our analysis demonstrates that the hybrid approach is more promising for developing general logical reasoning because (i) its reasoning chain is more interpretable, and (ii) it retains the capabilities and advantages of existing LLMs. To support future works using the hybrid approach, we propose a generalizable framework based on LLM-SS that is modular by design, model-agnostic, domain-agnostic, and requires little to no human input.

## 1. Introduction

State-of-the-art large language models (SOTA LLMs) steadily inch higher and higher on a wide array of benchmarks, from text summarization to code generation (Achiam et al., 2023). Nonetheless, LLMs continue to exhibit serious deficiencies in their ability to perform logical reasoning (Huang and Chang, 2022). Despite the gradual rise in accuracy in logical reasoning benchmarks, LLMs' fundamental problems have not been mitigated.

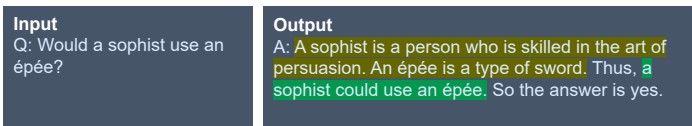

Figure 1: An example CoT output from a StrategyQA question in Wei et al. (2022)

Consider the example in Fig. 1, which shows a chain-of-thought (CoT) response to a question from the StrategyQA dataset (Geva et al., 2021), generated by Wei et al. (2022). In a CoT response, the model will generate intermediate reasoning steps before arriving at the final answer. The text highlighted in yellow shows the LLM's premises, i.e. the evidence used to support the conclusion, while the text highlighted in green shows the LLM's conclusion. This example highlights two primary flaws in LLMs: **(1)** the premises do not lead to the conclusion. Given the two premises, the final answer should have been "no"; **(2)** the conclusion may not have been derived from the premises at all. This problem is more subtle, for we naturally assume that if the conclusion does not follow from the premises, it implies the LLM has made a logical reasoning mistake. However, we cannot definitively state that the LLM inferred based on any and all of the premises, which are further explained below.

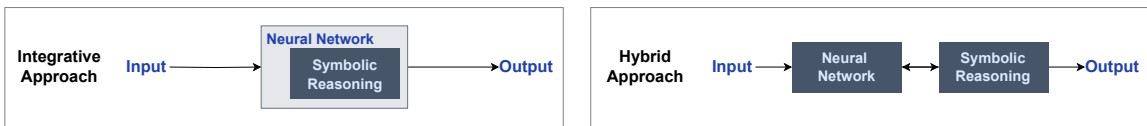

Figure 2: Integrative and hybrid neurosymbolic approaches for general logical reasoning

Both problems are manifestations of the limitations of Transformer-based architectures. Their probabilistic nature causes Flaw (1), while the lack of interpretability prevents Flaw (2) from being resolvable. In response to these challenges, neurosymbolic systems have recently regained prominence as an alternative to purely Transformer-based architectures; their deterministic and interpretable methods appear promising for enabling logical reasoning. Interpretability is defined as a model's ability to accurately demonstrate its flow of reasoning to arrive at an answer. As the name suggests, it aims to combine the best of both worlds: neural networks and symbolic reasoning. The former enables learning, creativity, and inductive reasoning, while the latter handles logical reasoning with symbolic rules and algorithms.

However, many recent neurosymbolic works (Badreddine et al., 2022; Riegel et al., 2020) are not generalizable as evidenced by how they are typically benchmarked on domain-specific tasks. For example, the CLUTRR benchmark, which only includes family relations, is commonly used (Sinha et al., 2019) to evaluate inference skills. This is because such models require a comprehensive list of task-specific axioms/formulas to be defined before training; additional details and examples are provided in Section A. Since it is unfeasible to manually define logical rules for hundreds of expansive topics, these models cannot be applied to domain-agnostic benchmarks like MMLU (Hendrycks et al., 2020), FOLIO (Han et al., 2022), and StrategyQA (Geva et al., 2021); they cannot achieve the broad applicability of existing LLMs. As such, we focus on general logical reasoning in this paper, which we define as the ability to deductively reason in domain-agnostic tasks.

Inspired by the taxonomies in Kautz (2020) and Ciatto et al. (2024), we identify two main approaches to neurosymbolic AI for general logical reasoning, i.e. integrative and hybrid, which are illustrated in Fig. 2. The **integrative approach** modifies the neural network architecture to allow it to perform logical reasoning in a deterministic and interpretable way.

On the other hand, the **hybrid approach** sidesteps the limitations of traditional neural networks by coupling them with external symbolic solvers. Recent literature found success with both approaches for domain-specific benchmarks like CLUTRR and StepGame (Shi et al., 2022), but their use in general logical reasoning is still understudied. The Related Works section can be found in Appendix A.

To evaluate the merits of both approaches, we develop a best-in-class model for each approach and compare their strengths and weaknesses. For the integrative approach, we create a novel neural network that consists solely of differentiable logic gates, which we refer to as Logic Neural Network (LNN). It can deterministically represent any and all laws of propositional calculus. For example, the statement "If $a$ and $b$, then $c$ is true." can be represented by an AND logic gate ($a \wedge b$). Moreover, it is interpretable. Once a specific neuron has chosen a logic gate, one can precisely interpret the argument form used. We experimentally validate that LNN is comparable or better than the current SOTA integrative model using both synthetic and real-world datasets.

For the hybrid approach, we introduce LLM-SS, a framework that combines an LLM and a symbolic solver. It is, by design, model-agnostic, domain-agnostic, and requires little human input. Broadly speaking, in the case of question-answering (QA) tasks, the LLM is responsible for generating natural language premises for the question, and then translating them into logical form. Afterward, the logical form is fed to the symbolic solver, which outputs the final conclusion using deductive reasoning. LLM-SS achieved higher or similar performance and lower error rates on domain-agnostic QA tasks compared to other models using the hybrid approach through the use of several novel techniques.

To evaluate which approach holds more potential for general logical reasoning, we formulate our comparison based on the following criteria: (i) ability to reason symbolically, (ii) interpretability of reasoning chain, and (iii) retention of LLM abilities. We find that while both integrative and hybrid approaches are able to reason symbolically, the former's interpretability decreases when model size increases and the former loses much of the capabilities of existing LLMs, such as knowledge retrieval and generalization. LNN and the integrative approach as a whole suffer from theoretical limitations that limit their potential for tackling general logical reasoning. Given these factors, we contend that the hybrid approach is more promising. Finally, we propose a neurosymbolic framework based on LLM-SS to support future works in this direction.

## 2. Model Architectures

### 2.1. Logic Neural Network (LNN)

Representing the **integrative approach**, LNN is a regular neural network with an adapted logic gate formula for each neuron, which builds upon the Logic Gate Network (Petersen et al., 2022). Every 2 neurons in a layer connect to a randomly chosen neuron in the subsequent layer. Each neuron has a choice of 16 distinct logic gates, as shown in Table 1. This is because, given 2 binary inputs, there are 4 unique input combinations. Since the output is also binary, there are 16 unique output combinations, each corresponding to a logic gate. 2 neurons are connected instead of 3 or more because the latter's implementation is considerably more complicated, yet does not exhibit stronger performance empirically given a comparable number of parameters (Petersen et al., 2022; Benamira et al.).

Table 1: List of logic gates with their corresponding relaxation formulas and outputs values given input neurons $a$ and $b$. This table is derived from Petersen et al. (2022).

| Logic Gate | Real-Valued Logic | $A{=}0, B{=}0$ | $A{=}0, B{=}1$ | $A{=}1, B{=}0$ | $A{=}1, B{=}1$ |
|---|---|---|---|---|---|
| False | $0$ | 0 | 0 | 0 | 0 |
| $A \wedge B$ | $A \cdot B$ | 0 | 0 | 0 | 1 |
| $\neg(A \Rightarrow B)$ | $A - AB$ | 0 | 0 | 1 | 0 |
| $A$ | $A$ | 0 | 0 | 1 | 1 |
| $\neg(A \Leftarrow B)$ | $B - AB$ | 0 | 1 | 0 | 0 |
| $B$ | $B$ | 0 | 1 | 0 | 1 |
| $A \oplus B$ | $A + B - 2AB$ | 0 | 1 | 1 | 0 |
| $A \vee B$ | $A + B - AB$ | 0 | 1 | 1 | 1 |
| $\neg(A \vee B)$ | $1 - (A + B - AB)$ | 1 | 0 | 0 | 0 |
| $\neg(A \oplus B)$ | $1 - (A + B - 2AB)$ | 1 | 0 | 0 | 1 |
| $\neg B$ | $1 - B$ | 1 | 0 | 1 | 0 |
| $A \Leftarrow B$ | $1 - B + AB$ | 1 | 0 | 1 | 1 |
| $\neg A$ | $1 - A$ | 1 | 1 | 0 | 0 |
| $A \Rightarrow B$ | $1 - A + AB$ | 1 | 1 | 0 | 1 |
| $\neg(A \wedge B)$ | $1 - AB$ | 1 | 1 | 1 | 0 |
| True | $1$ | 1 | 1 | 1 | 1 |

However, logic gates are discrete, with values of either 0 or 1, and are therefore non-differentiable. To enable gradient descent, each logic gate is relaxed into a continuous graph with 0 and 1 at each boundary, also known as real-valued logic. For example, the AND logic gate can be represented by $a \cdot b$ while the OR logic gate is represented by $a + b - a \cdot b$, based on probabilistic T-norm and T-conorm respectively (van Krieken et al., 2022). Notice that all formulas in Table 1 require at most inputs $a$, $b$, and $a \cdot b$, where $a$ and $b$ are the values of the input neurons. We can therefore further generalize the formulas into Eq. 1, where $a$ and $b$ are the input neurons, $c$ is the output neuron, and $w_{1-4}$ are trainable weights. The sigmoid function $\sigma$ ensures that output $c$ ranges from 0 to 1 during training. Each neuron there uses Eq. 1 during training.

$$c = \sigma(w_1 a + w_2 b + w_3(a \cdot b) + w_4) \tag{1}$$

At inference time, Eq. 1 is then translated into a discrete logic gate for each neuron. Each output $o_i$ of the 4 input possibilities is discretized into 0 or 1, such that when $o_i > 0.5$, it is assigned a value of 1. Based on the 4 outputs, the corresponding logic gate is determined. All reported results are based on accuracies from the discretized version.

### 2.2. LLM-Symbolic Solver (LLM-SS)

Representing the **hybrid approach**, LLM-SS is a multi-stage system, as illustrated in Fig. 3. In **Stage 1**, an LLM is prompted to generate natural language premises for an input question. Two types of propositional logic premises are allowed: (1) declarative sentences, which are statements that have truth values and no connectives, e.g. "A spider has 8 legs.",

and (2) conditional sentences, or if-else statements, e.g. "If an animal has 6 legs, it is not a spider." (Pospesel, 1974). In other words, this is similar to CoT (Wei et al., 2022), except the final answer is not generated and only two types of sentences are encouraged.

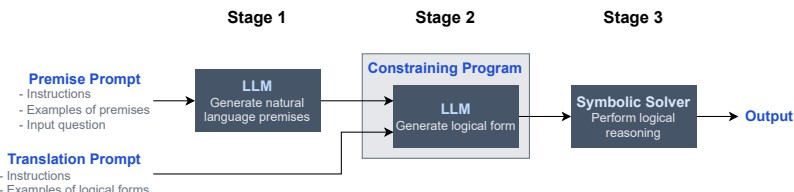

Figure 3: Architecture of LLM-SS model

In **Stage 2**, an LLM translates the natural language premises into logical forms, essentially a semantic parsing task. However, the few-shot examples cannot cover all forms of knowledge representations exhibited by uncontrolled natural language, which includes a multitude of vocabulary and grammatical structures, often leading to invalid logical forms. In fact, Lyu et al. (2023) find that syntax errors and infinite loops (a subset of syntax errors) account for 52.9% of all errors made by OpenAI Codex on the StrategyQA dataset. To tackle this issue, we develop a *novel constraining program* for the LLM. Specifically, based on constraints introduced by the formal grammar of the logical form syntax, the program masks the logits of tokens that violate these constraints. For example, a grammar may dictate that "=" must be followed by *True* or *False*. In this case, assuming the subsequent token is "=", all tokens besides those two will be masked.

In **Stage 3**, a symbolic solver receives the logical forms as input and then deductively derives the answer. To ensure determinism and interoperability, we turn to answer set programming (ASP) (Lifschitz, 2019), a form of logic programming (Lloyd, 2012). Logic programming represents natural language sentences as logical forms, which allow the logical relationships between entities to be unambiguously parsed by the program, unlike natural language sentences which may have multiple interpretations. ASP is a subset of logical programming, which focuses on solving search problems; finding the truth value of a statement based on a set of premises is one such problem. Clingo (Gebser et al., 2019), an ASP, is used in our experiments (which is further explained in Appendix D).

Importantly, so long as the premises and ASP code are accurate, the final output is necessarily true due to the deterministic nature of Clingo. This is unlike traditional CoT (Wei et al., 2022), where the final output may not be consistent with its premises, as highlighted in Figure 1.

## 3. Experimental Setup

We benchmark LNN and LLM-SS against other methods with integrative and hybrid approaches respectively. Note that different benchmarks are used for each approach. Currently, the integrative and hybrid approaches tackle completely different task domains; there is no common benchmark available to directly compare the approaches. Nonetheless, this experimental setup is sufficient to evaluate the strengths and weaknesses of each approach, which we further justify in Appendix C.

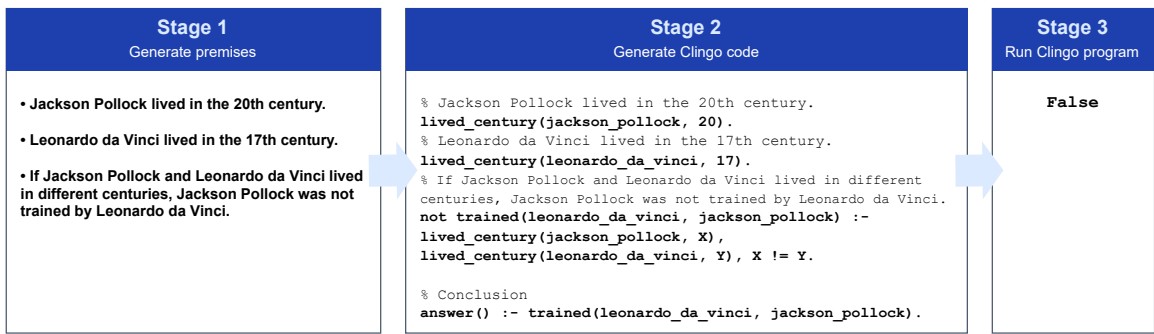

Figure 5: LLM-SS's response to "Was Jackson Pollock trained by Leonardo da Vinci?"

## 3.1. Logic Neural Network (LNN)

Our experiments for LNN aim to understand whether (i) the neurons are able to converge on the appropriate logic gate, and (ii) its performance on existing benchmarks. Both objectives are empirically studied in Experiments 1 and 2 respectively. This will help us understand the integrative approach's pros and cons, which we discuss in Section 5.1

For **Experiment 1**, we create a simple synthetic dataset where the model must identify the appropriate logic gate when provided with 4 unique sets of neuron $a$ and $b$ values and their corresponding output. For example, with reference to Table 1, when provided the outputs 0, 0, 0, 1 for their corresponding $a$ and $b$ values, the model is expected to identify the $A \wedge B$ logic gate.

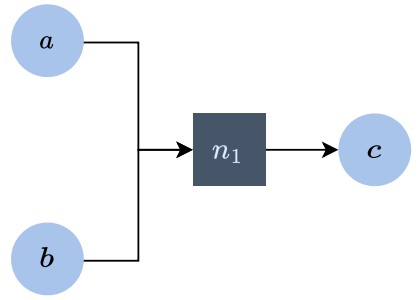

Figure 4: Experiment 1: Model Architecture of LNN & LGN

We benchmark LNN against the SOTA Logic Gate Network (LGN) model (Petersen et al., 2022) based on (i) whether it converges on the correct logic gate and (ii) how many iterations it takes. Models like the IBM Logical Neural Network and the Logic Tensor Network are excluded because they presuppose that the model already knows the logic gates used before training. This condition does not hold for the aforementioned task, thus making it beyond these models' existing capabilities. The architecture of both models is shown in Figure 4: a single-layer model containing a neuron with the respective formulas of LNN and LGN. When the neuron's discretized version chooses the appropriate logic gate, it is considered to have converged.

For **Experiment 2**, we utilize the Adult Census and Breast Cancer datasets (Asuncion et al., 2007), which are classification tasks containing 48842 and 286 instances respectively. LNN is benchmarked against LGN and a multi-layer perception (MLP). The model architecture and hyperparameters of LGN and MLP can be found in Appendix E.

### 3.2. LLM-Symbolic Solver (LLM-SS)

The task chosen is the StrategyQA dataset (Geva et al., 2021), where the model must infer the appropriate premises based on a domain-agnostic question and reason about those premises. Fig. 5 shows how LLM-SS will answer the question "Was Jackson Pollock trained by Leonardo da Vinci?". The model must first infer the appropriate premises in Stage 1, followed by their corresponding Clingo representations in Stage 2, which is ultimately used to generate the final answer using the Clingo program. Beyond testing for general logical reasoning, StrategyQA requires the advantages of existing LLMs: (i) learning, storing, and retrieval of knowledge, which is evaluated by whether models have the facts needed for the question from their training data, and (ii) inductive reasoning, which is evaluated by whether models can select the relevant facts.

We compare LLM-SS to (i) traditional CoT, (ii) Faithful-CoT, and (iii) an unconstrained LLM-SS, where the constraining software in Stage 2 is removed. LLM-SS uses Llama2-7B (Touvron et al., 2023) in Stage 1 and CodeQwen1.5-7B (Bai et al., 2023) in Stage 2; unconstrained LLM-SS and the traditional CoT model uses Llama2-7B; Faithful-CoT uses GPT-4. All models use Clingo as their symbolic solver, except Faithful-CoT, which uses Prolog, an alternative logic programming language. Few-shot prompts are executed with four examples due to Llama2-7B's shorter context length, while Faithful-CoT uses six. As for metrics, other than accuracy, we also measure the error rate, which is the percentage of questions with no answers generated. This happens when the CoT model does not produce a *yes* or *no* answer, or when the ASP code has a syntax error.

## 4. Results

### 4.1. Logic Neural Network (LNN)

The results of **Experiment 1** are shown in Table 2. LNN's accuracy of 100% serves as a simple, foundational check that Eq. 1 can converge on the correct logic gate for all 16 options before moving to the more complex Experiment 2. Moreover, while LGN also achieves 100% accuracy, it converges almost 3 times slower than LNN, suggesting the latter's relaxation formula may be more optimal for training.

Table 2: Results of Experiment 1

|  | Accuracy | Avg. Iterations Needed |
|---|---|---|
| Logic Gate Network | 1.00 | 163.4 |
| **Logic Neural Network** | **1.00** | **63.9** |

As for **Experiment 2**, the results are shown in Table 3. LNN achieves the highest accuracy (78.6%) out of the 3 models for the Breast Cancer dataset. The Adult Census, on the other hand, saw comparable results for all models, with MLP being marginally better (84.9%) than the rest. These results indicate that LNN outperforms LGN on smaller datasets due to the former's stronger convergence abilities but underperforms on larger datasets. While further research is needed to understand and refine LNN's scope and applicability, this lies beyond the scope of our work, as it is not essential for the analysis presented in Section 5.1.

Table 3: Results of Experiment 2

|  | Breast Cancer | | Adult | | Interpretable |
|---|---|---|---|---|---|
|  | Accuracy | Space | Accuracy | Space |  |
| Multi-Layer Perceptron | 0.753 | 1.4KB | **0.849** | 15KB | ✗ |
| Logic Gate Network | 0.761 | 320B | 0.848 | 640B | ✓ |
| **Logic Neural Network** | **0.786** | 320B | 0.848 | 640B | ✓ |

## 4.2. LLM-Symbolic Solver (LLM-SS)

As shown in Table 4, LLM-SS has a significantly higher accuracy and lower error rate compared to its unconstrained counterpart. It is evident that constraining LLM generation to enforce the syntax of Clingo leads to fewer errors during code execution, thereby increasing LLM-SS's accuracy. This is further highlighted when benchmarked against Faithful-CoT, which suggests a smaller constrained model, e.g., Llama2-7B, can perform better than a larger unconstrained model, e.g., GPT-4. However, while LLM-SS is interpretable, it still lags behind the traditional CoT model in terms of accuracy.

Table 4: Results of LLM-SS experiment

|  | Model | Accuracy | Error Rate (%) | Interpretable |
|---|---|---|---|---|
| CoT | Llama2-7B | **60.6** | **0.6** | ✗ |
| Faithful-CoT | GPT-4 | 54.0[1] | - | ✓ |
| LLM-SS (Unconstrained) | Llama2-7B | 48.5 | 17.8 | ✓ |
| **LLM-SS** | Llama2-7B | **54.5** | **1.5** | ✓ |

The main bottleneck of LLM-SS's accuracy can be straightforwardly deduced. The process in which the CoT model gathers facts is identical to Stage 1 of LLM-SS. Stage 3 of LLM-SS is a deterministic execution of Clingo, so it cannot be blamed for any errors. Thus, Stages 1 and 3 cannot explain the gap in accuracy between the two models. Thus, Stage 2 is the cause, specifically, the translation from natural language sentences to code. McGinness and Baumgartner (2024) categorizes translation errors into syntactic and semantic errors. Syntactic errors are defined as errors in the logical form that prevent parsing, while semantic errors are defined as logical forms that falsely represent their corresponding sentence despite being parsable. Given that the syntactic error rate for LLM-SS is 1.5%, it is evident that semantic errors are mostly responsible. Semantic error manifests in several ways: first, the naming convention between premises is sometimes inconsistent. For example, one premise may say "1519" while another may say "16th century", thus making the use of math operators to compare between them impossible. Second, the code translation may also be nonsensical, such as the usage of words and phrases that do not even appear in the premise.

---

1. This result is reported in https://github.com/veronica320/Faithful-COT

## 5. Discussion

### 5.1. Comparison of Integrative & Hybrid Approaches

Is the integrative or hybrid approach more promising for developing general logical reasoning without sacrificing the capabilities of existing LLMs? Given the strong performance of LNN and LLM-SS against SOTA models, they serve as case studies to answer the aforementioned question. We compare the approaches using the following criteria: (i) ability to reason symbolically, (ii) interpretability of reasoning chain, and (ii) retention of LLM abilities, i.e. whether the model can still preserve the advantages of existing LLMs, such as memorization and generalization. Table 5 summarizes our analysis.

Table 5: Comparison of integrative and hybrid approaches

|  | Symbolic Reasoning | Interpretability | LLM Abilities |
|---|---|---|---|
| Integrative | ✓ | ∼ | ✗ |
| Hybrid | ✓ | ✓ | ✓ |

**Criteria 1: Symbolic Reasoning.** LNN is able to logically reason, albeit in a limited fashion. Specifically, it is restricted by the connections pre-formed between neurons of one layer to the next: a pre-formed connection may not always accurately represent a given logical argument. Models should be allowed to learn the most optimal connections. Gumbel-Max Equation Learner Networks (Chen, 2020) uses Gumbel-Softmax to learn which outputs of the previous layer should be the input of the next layer. However, each layer's arithmetic operations are predefined. Since combining both flexible logical formulas and flexible connections has not been achieved but appears plausible, we argue that the integrative approach is capable of symbolic reasoning.

The hybrid approach, on the other hand, is capable of symbolic reasoning due to the use of symbolic programs. While Stage 2 currently hinders reasoning due to its inadequate translation abilities, this can be addressed by designing improved natural language-to-Clingo code translation models. Future works can draw inspiration from semantic parsing tasks like Abstract Meaning Representation (AMR) (Knight et al., 2021). Effective methods for AMR parsing include structure-aware transition-based approaches with pre-trained language models (Zhou et al., 2021) and model ensembling (Lee et al., 2021).

**Criteria 2: Interpretability.** Both approaches contain some degree of interpretability given their explicit use of symbolic reasoning: the logic gates used by LNN and the Clingo code used by LLM-SS are readily accessible. However, the former's interpretability scales poorly. Consider the LNN implemented for the Breast Cancer dataset which contains $5 \times 128 = 640$ neurons. While one can identify the logic gate of each neuron, a human cannot interpret a line of reasoning containing 640 logic gates. Therefore, LNN is only realistically interpretable when restricted to just a few logic gates. From a human perspective, as the number of neurons increases, LNN's interpretability becomes no different from an LLM.

For the hybrid approach, Lyu et al. (2023) point out that while the Problem-Solving stage in Faithful-CoT (or Stage 3 in LLM-SS) is transparent and interpretable, the Translation Stage (or Stages 1 and 2 in LLM-SS) is still opaque. One cannot interpret how premises and logical form translations are generated. However, we argue that this is not a fundamen-

tal issue. Consider an analogy to the human brain: a human's argument is deemed logical so long as the premises retrieved from memory lead to a conclusion. The criteria for an interpretable line of reasoning do not require transparency into how our brain retrieved the premises in the first place. Similarly, for LLM-SS, Stages 1 and 2 do not necessarily need to be deterministic and interpretable for logical reasoning to occur.

**Criteria 3: Retention of LLM Abilities.** Transformer-based architectures excel at inductive reasoning and the learning, storage, and retrieval of knowledge. To compete against existing LLMs in general logical reasoning, neurosymbolic models must retain these capabilities. The integrative approach replaces the Transformers architecture with a logic-based one, thereby entirely stripping the model of its LLM capabilities. Moreover, attempting to attain the advantages of LLMs conflicts with the reasoning capabilities of integrative models: when LNN is scaled to millions of neurons, it is no longer interpretable.

The hybrid approach, on the other hand, has no such issues. In LLM-SS, Stage 1 still uses a Transformer-based architecture, i.e. Llama2-7B, thereby retaining the capabilities of existing LLMs, while Stage 3 uses Clingo to perform symbolic reasoning. This separation of responsibilities allows LLM-SS to achieve the best of both worlds.

To summarize, the hybrid approach is superior to the integrative approach with regard to interpretability and retention of LLM abilities. We therefore posit that the hybrid approach is more promising for developing general logical reasoning.

### 5.2. LLM-SS: A framework for the hybrid approach

The framework exemplified by LLM-SS in Fig. 3 is generalizable. By segmenting LLM-SS into three distinct stages, it becomes a modular framework, where each stage can be independently tested, updated, and improved. Specifically, the choice of models for each stage is also flexible: alternative LLMs like GPT-4 and Gemini 1.5 can be used in Stage 1, semantic parsing models in Stage 2, and other symbolic solvers like Datalog and Python in Stage 3. Most neurosymbolic models are tailored to solve domain-specific tasks by using domain-specific knowledge modules and manually translated logical forms. In contrast, LLM-SS is designed for general logical reasoning, ensuring its viability across a variety of tasks, regardless of domain. To apply this framework to a new task, only in-context learning is required as human input.

To improve this framework, future directions include the (i) optimization of prompts and choice of models, (ii) mitigation of semantic errors in Stage 2, (iii) use of alternative logical forms in Stage 2 and 3, such as knowledge graphs (Pan et al., 2024), and (iv) evaluation of LLM-SS's effectiveness in other domain-agnostic tasks.

## 6. Conclusion

Our study finds that the hybrid approach to neurosymbolic AI is more promising than the integrative approach for developing general logical reasoning without sacrificing the capabilities of existing LLMs. To aid our analysis, we introduced the Logic Neural Network (LNN) and LLM-Symbolic Solver (LLM-SS) which serve as case studies and representatives for the integrative and hybrid approaches respectively. Finally, to support future research in neurosymbolic AI, we propose LLM-SS as a modular, model-agnostic, and domain-agnostic framework.

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

## Appendix A.  Related Works

With regards to the integrative approach, Garcez and Lamb (2023) reviewed several systems where symbolic reasoning is contained within the neural network. Most notably, Logic Tensor Network (Badreddine et al., 2022) is designed to learn new predicates via a deep neural network while satisfying a first-order logic knowledge base. Daniele and Serafini (2019), Fischer et al. (2019), and Manhaeve et al. (2018) present models with similar principles. Logical Neural Network (Riegel et al., 2020), developed by IBM, creates a 1-on-1 correspondence between each neuron and a logic gate, a similar concept to our LNN.

However, the primary difference between the two aforementioned systems and LNN is that the former requires Real Logic axioms/formulas specific to the given task to be manually defined prior to training. The model subsequently learns the weights with respect to the axioms provided. LNN, on the other hand, consists of the same 6 logic gates for each neuron across any given task. It learns each neuron's optimal logic gate during training, thus dynamically constructing the best-performing axioms/formulas. This is a significant disadvantage for the Logic Tensor Network and the IBM Logical Neural Network because it limits their scope to problems with known, well-defined logical rules, thereby diminishing their usefulness in real-world applications. For example, IBM's Logical Neural Network is tested on the Lehigh University Benchmark (LUBM) (Guo et al., 2005), which contains predefined OWL axioms.

To the best of our knowledge, the only model architecture that allows for the choice of a logic gate is the Logic Gate Network (LGN) (Petersen et al., 2022). Its fundamental principle is the same as LNN: allow neural networks to choose between logic gates through differentiation by continuously relaxing them. However, LNN and LGN differ in how various logic gates are combined. This paper creates a distinct relaxation formula for each logic gate and then uses a categorical probability distribution to create a weighted average across 16 logic gates. However, LGN's primary objective was to decrease inference time for computer vision tasks by using the highest probability logic gate for each neuron. Hence, its practical ability to converge to the appropriate logic gate for reasoning tasks remains unexamined.

On the other hand, the hybrid approach has seen SOTA results on domain-specific reasoning tasks, but few have applied this approach to domain-agnostic tasks. Yang et al. (2023) applied GPT-3 and Clingo to the benchmark datasets: bABI, StepGame, CLUTRR, and gSCAN. However, the scope of its application is limited. For example, CLUTRR only involves the inference of family relationships. Moreover, the authors manually wrote an ASP knowledge module for each task, e.g. the one for CLUTRR contains an exhaustive list of family relationships. Therefore, Yang's architecture is not applicable to general QA tasks. Deepmind's AlphaGeometry (Trinh et al., 2024) combines an LLM with the symbolic engine DD+AR, which contains geometric rules. Similarly, its scope is limited to geometry problems. Moreover, these natural language geometry questions are manually translated into a domain-specific form in order for DD+AR to understand the problem. This is again unrealistic for general QA tasks. Models with similar frameworks, and therefore similar limitations include Silver et al. (2016), McGinness and Baumgartner (2024), Zhang et al. (2023), and Dai et al. (2019).

To our knowledge, only the Faithful-CoT model (Lyu et al., 2023) attempts to combine LLMs with symbolic engines for a wide range of tasks, without knowledge modules or

manual translation of problems. However, using GPT-4 and Prolog, an alternative to Clingo, it achieved a mere 54% accuracy rate on the StrategyQA benchmark (Geva et al., 2021).

## Appendix B. Are LNN and LLM-SS Generalizable Representatives?

This study compares one method only from each approach, which therefore begs the question: are LNN and LLM-SS reasonable representatives of each approach? In other words, can their results and analyses be extrapolated to form conclusions about each approach as a whole? Our conviction is that both methods' analyses can be generalized as a comparison between both approaches, justified by the key characteristics shared by the class of models within each approach.

Referencing Appendix A, the integrative approach is characterized by the embedding of logic gates or axioms within the neural network, which are manipulated to various degrees during training via gradient descent. This defining property of integrative models is undoubtedly represented by LNN. The hybrid approach, on the other hand, is a system characterized by two distinct parts: a probabilistic neural network and a deterministic symbolic solver. This is similarly well-represented by LLM-SS. Importantly, our analysis is contingent upon the aforementioned universal key characteristics, rather than any model-specific implementation method. We therefore expect our conclusions in Section 5.1 to hold regardless of the choice of model representatives.

## Appendix C. Explaining the Experiment Design

Why are the integrative and hybrid approaches evaluated using different benchmarks, preventing them from being directly comparable? This is due to the current limitations of both approaches. On one hand, the integrative approach has not yet been scaled to accommodate natural language datasets, and thus can only be trained on classification tasks with limited features. The hybrid approach, on the other hand, specializes in natural language datasets, given its use of LLMs; it cannot reliably train on datasets like Adult Census without defeating its purpose. Thus, their incompatible task domains prevent a direct comparison.

Nonetheless, we contend that this experimental setup provides sufficient empirical support for our conclusions. Ultimately, we are interested in studying which approach is more interpretable and which better retains the advantages of existing LLMs; a direct performance comparison is secondary. Our experiments adequately study these questions.

## Appendix D. Clingo as Part of LLM-SS

Below are examples of how natural language sentences, both declarative and conditional, can be represented in Clingo. Note the ability to use mathematical expressions in the last example.

- [Declarative] Sam has a cow.
  ```
  no_of_cows_owned(sam, 1).
  ```

- [Conditional] If Sam owns a cow, Sam is a farmer.
  ```
  farmer(sam) :- no_of_cows_owned(sam, 1).
  ```

- [Conditional] If Sam owns more than five cows, then he is rich.
  `rich(sam) :- no_of_cows_owned(sam, Number), Number >5.`

This explains the architectural choices in the earlier stages. Restricting the premises generated in Stage 1 to declarative and conditional forms allows a simpler and therefore more error-free translation into logical form. Moreover, using LLM constraining software in Stage 2 helps ensure the LLM adheres to Clingo's strict syntax rules.

**Choice of Clingo.** In theory, any ASP or logic programming language can be used as part of LLM-SS. However, our choice of Clingo over Prolog, another commonly used solver, is twofold. First, Clingo's syntax is more consistent with propositional logic in terms of semantic meaning. For example, negation in propositional logic ($\neg P$) and in Clingo (`not P`) both mean $P$ is false. However, negation in Prolog (`\+ P`) means $P$ cannot be proven, which is distinctly different. This makes the natural language-to-code translation unnecessarily complex. Second, as a secondary consideration, Clingo's syntax is simpler than Prolog's and may therefore be less prone to random errors, e.g., for negation (Prolog's `\+` vs. Clingo's `not`). We recommend future works to select symbolic solvers based on (i) alignment with the chosen logic system and (ii) ease of parsing and generation by LLMs.

## Appendix E. Model Architecture and Hyperparameters

For LNN's Experiment 1, both models are trained with 1000 iterations, a learning rate of 0.01, and using the Adam optimizer.

For LNN's Experiment 2, the model architectures and hyperparameters are determined by the recommended settings in Petersen et al. (2022), which are as shown in Table 6. Importantly, model sizes are kept roughly equivalent in the interest of fairness. All models are trained up to 200 epochs at a batch size of 100. The MLPs are ReLU activated.

| Model | Space | Layers | Neurons/layer |
|---|---|---|---|
| **Breast Cancer** | | | |
| Logic Neural Network | 320B | 5 | 128 |
| Logic Gate Network | 320B | 5 | 128 |
| Multi-Layer Perceptron | 1.4KB | 2 | 8 |
| **Adult** | | | |
| Logic Neural Network | 640B | 5 | 256 |
| Logic Gate Network | 640B | 5 | 256 |
| Multi-Layer Perceptron | 15KB | 2 | 32 |

Table 6: Model architecture and hyperparameters for LNN's Experiment 2

