# OpenReview forum: "A Comparative Study of Neurosymbolic AI Approaches to Interpretable Logical Reasoning"
_nesyconf.org/NeSy/2025/Conference_Phase_2 — NeSy 2025 - Phase 2 Poster_

### Official Review · Reviewer_SC4D · 2025-07-03
**review of Submission20**

**Rating:** 6
**Confidence:** 4

**Review:**

## summary of the paper

The authors assert and consider two categories of approach to blending domain-agnostic general logical (symbolic, deductive) reasoning with deep learning:
1. an *integrative approach*, where symbolic reasoning is performed *within* neural network models
2. a *hybrid approach*, where symbolic reasoning is performed *externally* to neural network models, using a symbolic solver of some kind.

The central research question motivating the paper is which of the two approaches is more promising, with respect to three criteria:
* ability to reason symbolically
* interpretability of the reasoning chain
* retention of LLM capabilities.

To study this question, the authors introduce and propose a novel representative system for each of the two categories of approach.  For the *integrative approach*, they introduce a Logic Neural Network (LNN) that is a variation of the Logic Gate Network (LGN), introduced by others.
For the *hybrid approach*, they introduce an LLM-SS (LLM Symbolic Solver) model that appears to be a variation of the 'Faithful Chain-of-Thought' model, introduced by others.

The central task for LNNs/LGNs is to learn a mapping between the neurons of the model's architecture and logical operations (expressed as fuzzy logic tnorms and co-norms). There are 16 logical operator possibilities for each neuron. Performance is evaluated in terms of the accuracy of the mapping that was learned, and in terms of the accuracy of trained models on two different classification tasks (datasets).

The central task for an LLM-SS is to answer natural language questions with a Boolean response of True or False, in a manner that exhibits (as best as possible) sound symbolic, deductive reasoning. The performance of the LLM-SS is compared against two other SOTA LLM chain-of-thought models designed for this same logical reasoning task. Performance is evaluated in terms of accuracy with respect to answering the questions posed in the StrategyQA dataset.


## evaluation of the paper

**clarity / use of language / quality of writing**

The paper is written clearly and its use of language is fine. Its organisation is a tad unorthodox in that it relegates the 'Related work' section to an Appendix (presumably to help stay within the NeSy 2025 10-page limit on long papers).

**novelty / originality**

The paper's novelty / originality is reasonable. The novel feature of the LNN (relative to LGN, upon which LNN is based) is that the formulae for the 16 logical operations are expressed in a single formula involving a weighted sum of the 4 possible terms of an operation. The weights in the sum are learned by gradient descent, for each neuron. Novel features of the LLM-SS (relative to 'Faithful Chain-of-Thought') include 1) the use of Answer Set Programming for the symbolic solver, 2) constraining the Stage 2 LLM (with procedural code) to ensure that the ASP programs it generates are at least correct syntactically, if not semantically.

**impact / significance**

The paper's impact / significance is modest. Like the LGN, the LNN achieves 100% accuracy in its task of mapping neurons to logical operations.  This suggests the learning task itself is not difficult. With respect to the two classification tasks, LNN delivers the same accuracy as LGN on one dataset, and marginally better accuracy than LGN on the other dataset.  That is, LNN learns faster than LGN, but not much better. One of the two classification datasets is small and the other larger. Based on their results, the authors proceed to assert that LNN works better on small datasets and LGN on larger datasets. These generalised conclusions are not supported by the small sample size of test results (one small dataset, one large); so making them dimishes the paper's authority rather than strengthens it.

The LLM-SS system does not change the dial with respect to SOTA performance on the StrategyQA dataset. The LLM-SS performs less well than one of the two competing Chain-of-Thought models, and only very slightly better than the other Chain-of-Thought model --- and it's not at all clear that the small reported uptick in performance is statistically significant.


**overall quality**

There is an important aspect of the paper that I find problematic, however: its overall conception or, what the authors would refer to (I think) as, its *experimental setup*.  The paper purports to be a comparative study of two alternate approaches to building NeSy systems that can do logical deductive reasoning --- to see which is more promising. But the paper feels more like an awkward juxtapositioning of disparate systems than a meaningful comparison. The authors are clearly self-conscious about this very point. They try to argue the validity of the setup, but the reader is not convinced. (Shifting their argument into an Appendix does not help in this regard.) The representatitve systems of the two NeSy approaches (integrative and hybrid) --- ie the LNN and the LLM-SS --- are incompatible sytems that do incompatible things on incompatible data. There is little basis for comparison.  And when one of the three decision criteria is 'retention of LLM capabilities', the reader is not surprised when the paper's main conclusion is that the hybrid approach (and their LLM-SS system) is the more promising approach.

I think it is also possible to argue that the LNN (and LGN, for that matter) does not demonstrate symbolic logical reasoning being performed *within* a neural network at all. The LNN and its training feels to me to involve conventional subsymbolic learning only. The model uses gradient descent (subsymbolic learning) to learn real-valued parameters that collectively represent a particular mathematical function that deterministically maps inputs to outputs.  A forward-pass of this fixed mathematical function does not demonstrate the performance of symbolic, logical, deductive reasoning, in my view. But this is debatable, I suppose.

The paper's situation in this respect is unfortunate because the work done individually with the LNN, and with the LLM-SS, is interesting and substantial and has novelty. One can readily imagine a separate paper for each of the two approaches: 1) an integrative approach paper, that introduces LNN and compares it with LGN, and with IBM's Logical Neural Networks (Riegel et al.), and with perhaps LTN (logic tensor networks), and 2) a hybrid approach paper that focuses on LLM-based systems that use symbolic solvers.


**minor observations**

Consider defining the `CoT` (chain of thought) acronym. It's heavily used but never defined.

Section 3, 1st paragraph: These two opening paragraphs need work. A reference to an Appendix is undefined: I think it's meant to point at Appendix B. The 2nd paragraph is malformed and incomplete.

It's arguable that LTN (Logic Tensor Networks) does not satisfy the definition of the *integrative approach* asserted by the authors. LTN is not a type of neural network, it is a framework for training conventional neural networks. With LTN, the logic never goes inside a neural network --- it is always outside, in the knowledge base of logical constraints that forms the LTN loss function.

**Anonymity:**

Remain anonymous

---

### Official Review · Reviewer_7DNa · 2025-07-06
**Neurosymbolic Approaches to Interpretable General Logical Reasoning: A Comparative Study**

**Rating:** 6
**Confidence:** 4

**Review:**

The paper compares the integrative approach of neurosymbolic learning and reasoning with the hybrid approach of neurosymbolic learning and reasoning in order to answer the question which approach is more promising for implementing general logical reasoning. More concretely the paper compares Logic Neural Networks (LNN) where logic gates are implemented on the neuron level with LLM-Symbolic Solvers where an LLM generates premises and rules that are translated into a constraining logic representation that is solved by a symbolic solver (in an answer set programming style). The results show that both approaches allow logical reasoning, but the hybrid approach has advantages with respect to interpretability and the preservation of LLM abilities.

I think the paper is of interest for the neurosymbolic community. It addresses an interesting question about the future directions of realizing neurosymbolic learning and reasoning. Furthermore, it is in general appropriately written with only minor weaknesses.
Here are some questions that I have about the paper:

In Section 2.1 a special Logic Neural Network architecture is introduced without a clear view why this architecture is general enough to comprise a whole bunch of integrated architectures (Logic Tensor Networks, Vector Symbolic Representations, DeepProbLog etc.). In Appendix A there are some remarks that clarify the situation partially. Nevertheless, it is not clear, why the architecture described in Section 2.1 suffices to cover the integrative approach in general.

Conceptually, I am not completely happy with the mixture of fuzzy logic and probabilistic logic in the paper (page 4). Fuzzy Logic is modeling vagueness, probabilistic logic is modeling uncertainty. T-norms and T-conorms are defined for Fuzzy operators not for probabilistic logic. Van Krieken et al, 2022 is considering Fuzzy logic, not probabilistic logic (as stated in the paper) and although van Krieken et al. (2022) says something about DeepProbLog in section 11.3 the different theories should be clearer separated from each other.

Concerning the modeling of differentiable logics, why is the paper focusing on the product T-norm. Is it because the product T-norm is closer to probabilistic reasoning than the Gödel T-norm or the Lukasiewicz T-norm (or any other T-norm)? In any case, the paper should somehow specify why this particular product logic is used and not any other logic.

In the evaluation of the LLM-Symbolic Solvers, all models use Clingo as solver except Faithful-CoT which uses Prolog. Is there a difference in the performance of the two types of solvers given that Clingo is a solver designed for answer set programming and Prolog is a general theorem prover?

In the discussion section, the paper infers the superior appropriateness of the hybrid approach from the features “Symbolic Reasoning”, “Interpretability”, and “LLM Abilities” (page 9). To ask in a provocative style: Why do we need sections 1 to 4? A sequential hybrid model is in any case better suited for interpretability than an integrated model. Similarly, a sequential hybrid model will not lose any LLM abilities, which is less obvious for integrated approaches. That both approaches can model symbolic reasoning is clear from the literature. In total, the result is somehow trivial, although the modeling in Sections 1 to 4 are definitely interesting.

Pros:
- The paper is interesting to the NeSy community
- The paper is appropriately written
- It contains some interesting considerations about integrated and hybrid approaches towards neurosymbolic learning and reasoning.

Cons:
- Some conceptual issues should be clearer distinguished from each other
- Further explanations concerning the compatibility of solvers or the choice of underlying logic are needed
- The generality of the approach is not clear to me
- The features which are used for the main result namely to prioritize hybrid models over integrated approaches seems to be trivial.
- The paper seems to be preliminary.

Some minor issues:
Page 5: second last paragraph: Reference for the appendix is missing.
Page 5: Second last line: “This” -> “this”
Page 6: The naming of “Logical Neural Networks” and “Logic Neural Networks” is quite confusing given that they are incompatible with each other. Is it possible to clearer distinguish them from each other?
Page 6, last paragraph: the used datasets should be intuitively described somewhere. Else, it is hard for the reader to understand the type of problems that the NeSy architectures should solve.

**Anonymity:**

Remain anonymous

---

### Official Review · Reviewer_Nq36 · 2025-07-10
**-**

**Rating:** 5
**Confidence:** 3

**Review:**

The paper attempts to compare the "integrative" and "hybrid" approach to neuro-symbolic reasoning.

While this sounds like an interesting endeavour, the study seems to be at a very early stage. The paper introduces one model for each approach. The models have not been described very well and there is little discussion of their relationships to existing work.

The experiments look somewhat random to me. For the first approach, the first experiment evaluates how well the model can learn a logic gate with 4 inputs. There is little discussion of the relevance of this task. The second task is a classification task and one wonders why one would apply a neuro-symbolic system here. Interpretability is mentioned as a motivation, but there is not much evidence that there is really an advantage compared to existing interpretable models like decision trees and logistic regression or tree ensembles with their accompanying feature scores. The second approach is evaluated on the StrategyQA dataset, which looks like a more interesting experiment. However, I am not sure what to take away from the results, and the choice of baselines seems somewhat random.

It would be good to add more motviation, more discussion of related work, and to improve the description of the models, their motivation, their relationship to the state-of-the-art, and to improve the experiments by motivating the datasets and baselines better and by explaining how they are related to state-of-the-art experiments in the literature.

While this looks like an interesting research direction, I cannot recommend the paper for acceptance in its current state.

**Anonymity:**

Remain anonymous

---

### Official Review · Reviewer_xFFK · 2025-07-10
**Comparison of neuro-symbolic architectures**

**Rating:** 6
**Confidence:** 3

**Review:**

The paper is an overview of approaches that incorporate logic into neural networks. The authors identify two main types of approaches:
in integrative approaches symbolic reasoning is done within the neural net while in hybrid approaches it is done in parallel to the neural inference.
The paper adds to the domain by comparing SOTA approaches, and by doing so it highlights which approaches are more promising in terms of generalizing and logical reasoning.

One obvious weakness is that the two approaches are represented by one model each: LNN as an integrated model and LLM-SS as a hybrid model.
So the question is whether results can be extrapolated on the whole classes of models or we are still comparing individual models.
One the one hand, the architecture of LLM-SS is quite generic and can indeed act as a representative of the hybrid models class (but even this model's performance depends on the exact implementation of the constraining program and other components; btw the constraining program should be described in the Appendix).
On the other hand, LNN the integrative approach model, is essentially the implementation of Petersen et al. 2022, a very unique and very specific type of model.

The authors do compare both models with other systems which helps back up more general conclusions. But then again, while LLM-SS is stacked against several SOTA models in the same class, LNN is only compared to Petersen et al. 2022 (+- the same model) and to a simple MLP.

So my worry is that we are comparing only one particular type of model (LNN) with a class of models (hybrid).
Therefore, I would be more cautious in making conclusions that generalize by induction: the authors suggest that "hybrid approach to neurosymbolic AI is more promising than the integrative approach for developing general logical reasoning" while they only consider one type of integrated architecture.

All that being said, I still think that it is an insightful paper. It is well written and clearly structured, and while the two approaches cannot be compared directly since there is no common benchmark, they are being compared along 3 relevant criteria: ability to reason symbolically, interpretability, and the ability to retain LLM abilities.
For examples, the authors identified the main bottleneck of hybrid models that should be researched more thoroughly next: NL-to-formula translators.
Another interesting observation is that while the LNN model seems to integrate logical reasoning into neural nets in the most native way, its interpretability (which is one of the main reasons to have integrated formal reasoning) scales poorly, and this is an important factor.

Overall, I think this paper could be a nice addition to the conference and would benefit the readers.

Minor:
- page 5: "we justify in Appendix ??"
- page 5: "Thus, This experimental setup"
- page 8: "Semantic error manifests in several ways: First,"

**Anonymity:**

Remain anonymous